# SUSI: Semi-Structured Pruning for LLMs via Differentiable Subset Sampling

## Abstract

The rapid growth of large language models (LLMs) has driven the need for efficient post-training optimization techniques for reducing computational and memory demands while preserving performance. Semi-structured pruning, which enforces hardware-compatible sparsity patterns like N:M sparsity, offers a balanced approach for accelerating inference. In this study, we introduce SUSI[1] (Semi-structured prUning via Subset samplIng), a novel semi-structured pruning method that leverages the weighted reservoir and differentiable subset sampling to learn high-quality N:M sparsity masks with minimal computational cost. Compared to other learnable mask methods (i.e., MaskLLM), which increase parameter complexity, SUSI reduces trainable parameters by up to 1.5× for the 2:4 sparsity, enabling efficient deployment on hardware optimized for sparse computation. We evaluate SUSI on three OPT model variants (125M, 350M, and 1.3B parameters) using benchmarks including Wikitext-2 for perplexity and zero-shot NLP tasks (e.g., ARC, HellaSwag, PIQA, RACE, SciQ). SUSI outperforms baselines such as SparseGPT, Wanda, and MaskLLM in perplexity while maintaining competitive zero-shot accuracy across various benchmarks. These results establish SUSI as a robust and practical solution for compressing LLMs, facilitating efficient deployment in resource-constrained environments.

## 1 Introduction

With the rapid development of large language models (LLMs), post-training techniques have emerged as critical methodologies for optimizing model efficiency while preserving performance (Wan et al., 2024). Among these techniques, two primary approaches to network compression have gained prominence: model quantization (Egashira et al., 2024; Liu et al., 2025b) and network pruning (Cheng et al., 2024; Muñoz et al., 2025). While model quantization focuses on representing weights with reduced precision (e.g., 8-bit, 4-bit, or lower), pruning techniques aim to eliminate redundant parameters to accelerate inference while preserving task performance (Williams & Aletras, 2024). This study focuses on pruning techniques to develop sparse LLMs, thereby reducing memory footprint and enhancing inference speed.

Current post-training pruning methods can be categorized into three distinct approaches: (i) unstructured pruning, which removes individual weight parameters without regard to network architecture (Sun et al., 2024); (ii) structured pruning, which eliminates entire network components such as neurons, attention heads, or layers (Xia et al., 2024; Le et al., 2025); and (iii) Semi-structured pruning, which combines the flexibility of unstructured methods with the regularity of structured patterns (Fang et al., 2024; Huang et al., 2025). This research focuses on semi-structured pruning, as it efficiently removes redundant weights while enforcing regular sparsity patterns that are hardware-compatible and effective for acceleration. Specifically, semi-structured pruning strikes an optimal balance by keeping regular sparsity patterns (e.g., N:M sparsity (Hubara et al., 2021)), which is optimized for hardware. Modern approaches in this field are generally categorized into two types: i) *importance-based*: with several typical methods such as SparseGPT (Frantar & Alistarh, 2023) and Wanda (Sun et al., 2024) using a small dataset, typically a subset of the pretraining data, to approximate the knowledge encoded in the language model. They define an importance score for each weight (or group of weights) based on this dataset, which guides the pruning process. Importance

---

[1] https://anonymous.4open.science/r/susi-2E2C

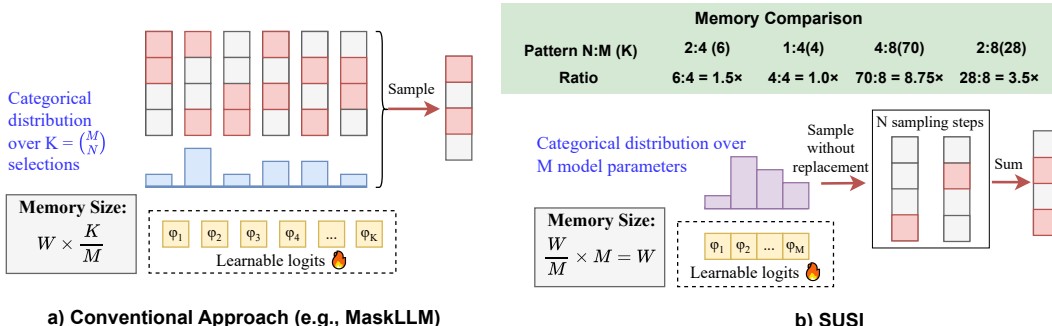

Figure 1: Learnable semi-structured N:M sparsity methods: a) modeling the mask selection process using a categorical distribution over feasible masks, and b) our proposed method by learning to sample subsets without replacement of model parameters. The proposed method is more memory efficient than previous works for most practical N:M sparsity patterns. The memory advantage becomes more pronounced as $M$ increases or when $N$ is around $M/2$.

scores may be based on weight magnitude, gradients, or the Hessian matrix. However, these criteria are often chosen heuristically, leading to potentially sub-optimal results. Additionally, the limited dataset may not adequately capture the model's rich knowledge; ii) *learnable masks*: focusing on the direct optimization of pruning masks through a retraining process. Recently, MaskLLM (Fang et al., 2024) proposed a novel method that models N:M sparsity patterns as learnable categorical distributions, employing Gumbel-Softmax sampling (Jang et al., 2017). This approach demonstrates robust pruning performance and strong generalization across diverse tasks. However, it introduces significant computational overhead due to an increased number of trainable parameters (Huang et al., 2025). Specifically, for a model with $W$ parameters under N:M sparsity, MaskLLM requires learning $\binom{M}{N} \times \frac{W}{M}$ parameters, which consistently equals or surpasses the original model parameter count, as illustrated in Figure 1(a). For instance, with the commonly utilized 2:4 semi-structured sparsity pattern, the number of parameters to be learned is $1.5 \times W$. This substantial parameter overhead poses considerable challenges during the training of large-scale language models.

To address this limitation, we propose an effective semi-structured pruning method, termed SUSI (Semi-structured prUning via Subset samplIng). SUSI systematically selects $N$ weights from each group of $M$ consecutive parameters, enabling the enforcement of N:M sparsity with minimal degradation in model accuracy. The main idea is to utilize Weighted Reservoir Sampling (WRS) (Efraimidis & Spirakis, 2006) as an efficient alternative for learning high-quality sparsity masks. WRS enables selective sampling of mask configurations based on importance weights, reducing computational overhead while maintaining the ability to identify effective N:M sparsity patterns. The proposed lightweight pruning mask learning technique significantly reduces the number of trainable parameters, thereby facilitating efficient deployment on hardware optimized for N:M sparsity, as depicted in Figure 1(b).

## 2 PRELIMINARIES

### 2.1 WEIGHTED RESERVOIR SAMPLING

Weighted Reservoir Sampling (WRS) (Efraimidis & Spirakis, 2006) is an extension of the Reservoir Sampling class of algorithms (Vitter, 1985), which aims to sample $K$ items from a set of $N$. In WRS, each item is assigned a non-negative weight, and items with larger weights compared to others are more likely to appear in the sampled subset. Given a population set $\mathcal{X} = \{x_1, x_2, \ldots, x_N\}$ with corresponding weights $\boldsymbol{w} = [w_1, w_2, \ldots, w_N]$, WRS produces an ordered subset $\mathcal{Y} = \{y_1, y_2, \ldots, y_K\}$, which is drawn from following distribution:

$$P_{\text{WRS}}(\mathcal{Y}|\boldsymbol{w}) = \frac{w_{y_1}}{W} \times \frac{w_{y_2}}{W - w_{y_1}} \times \ldots \times \frac{w_{y_K}}{W - \sum_{j=1}^{K-1} w_{y_j}} \tag{1}$$

where $W = \sum_{i=1}^{N} w_i$ is the total weight and $w_{y_i}$ is the weight of the corresponding item $y_i$. Sampling from the above distribution resembles the sampling without replacement process, where the probability of selecting a subset is proportional to the item weights.

## 2.2 GUMBEL-TOP-$K$ TRICK

Gumbel-Max (Gumbel, 1954) is a monotonic transformation of the WRS technique, a reparameterization trick to sample from a categorical distribution by perturbing the distribution's log-probabilities with Gumbel noise. Given a categorical distribution over $N$ items $\{x_1, \ldots, x_N\}$ parameterized by $N$ logit parameters $\boldsymbol{\phi} = [\phi_1, \ldots, \phi_N]$, the probability of an arbitrary item's selection is $\pi_i = \exp(\phi_i)/\sum_{j=1}^{N} \exp(\phi_j)$. The Gumbel-Max trick performs sampling from such a distribution by first generating random keys corresponding to each item via Gumbel perturbations:

$$\kappa_i = \phi_i + g_i, \quad g_i \overset{\text{i.i.d}}{\sim} \text{Gumbel}(0,1) \tag{2}$$

where $g_i$s are noise independently drawn from the $\text{Gumbel}(0,1)$ distribution. Finally, the output of this sampling process is achieved by taking the item $x_j$ having the largest key $\kappa_j$. The index $j$ is the output of taking $\text{argmax}$ over key values ($j = \text{argmax}_i \kappa_i$).

Gumbel-Top-$K$ is a generalization of the Gumbel-Max trick, where instead of selecting the item with the largest random key, the top-$K$ items with the highest keys are selected (Xie & Ermon, 2019). This corresponds to sampling $K$ items without replacement from a categorical distribution over $N$ items. By relaxing the $\text{argtop}_K$ operator using successive softmaxes (Plötz & Roth, 2018), this sampling process becomes differentiable, thereby allowing for learning with backpropagation.

To sample a subset of $K$ items with the Gumbel-Top-$K$ trick, logits are first independently perturbed with Gumbel noise to create random keys $\kappa_i$, similar to the Gumbel-Max trick. Sequentially, a chain of $\text{softmax}$ is applied to produce approximated one-hot representations of selected items. Let $\boldsymbol{\alpha}^{(k)} = [\alpha_1, \ldots, \alpha_N]$ denote adjusted keys at the sampling step $k$. These adjusted keys are defined recursively as follows:

$$\boldsymbol{\alpha}^{(1)} := [\kappa_1, \ldots, \kappa_N]; \quad \boldsymbol{\alpha}^{(k)} := \boldsymbol{\alpha}^{(k-1)} + \log(1 - \boldsymbol{\mu}^{(k-1)}) \tag{3}$$

where $\boldsymbol{\mu}^{(k-1)} = [\mu_1^{(k-1)}, \ldots, \mu_N^{(k-1)}]$ is the one-hot approximation indicating the item selected at the previous sampling step. This representation is achieved by applying $\text{softmax}$ over adjusted keys at the sampling step $k-1$ with a pre-defined temperature $\tau$:

$$\mu_i^{(k-1)} = \frac{\exp(\alpha_i^{(k-1)}/\tau)}{\sum_{j=1}^{N} \exp(\alpha_j^{(k-1)}/\tau)} \tag{4}$$

After applying $\text{softmaxes}$ $K$ times, we attain an ordered subset of $K$ approximated one-hot representing selected items $\mathcal{S} = \{\boldsymbol{\mu}^{(1)}, \ldots, \boldsymbol{\mu}^{(K)}\}$. The sum of elements in this subset yields a soft $K$-hot vector, and the mapping from the logits $\phi_i$ to this vector is differentiable, enabling usage of gradient-based optimization methods.

# 3 METHODOLOGY

## 3.1 PROBLEM STATEMENT

The problem of finding the optimal N:M sparsity can be formulated as selecting, for each group of $M$ consecutive parameters, a binary mask of length $M$ with exactly $N$ non-zero entries that minimizes the loss on a calibration set. Let $G$ denote the number of weight groups, $\mathbf{W} = \{\mathbf{w}_1, \ldots, \mathbf{w}_G\}$ the corresponding weight groups, and $\mathbf{M} = \{\mathbf{m}_1, \ldots, \mathbf{m}_G\}$ the associated binary masks. The optimization problem is then defined as follows:

$$\mathbf{M}^* = \underset{\mathbf{M}}{\text{argmin}} \, \mathcal{L}_{\text{CE}}(\mathcal{D}; \mathbf{W} \odot \mathbf{M}) \tag{5}$$

where $\mathcal{L}_{\text{CE}}$ is the cross-entropy loss for language modeling, $\mathcal{D}$ denotes the calibration set, and $\odot$ represents the element-wise product between each weight group and its corresponding binary mask.

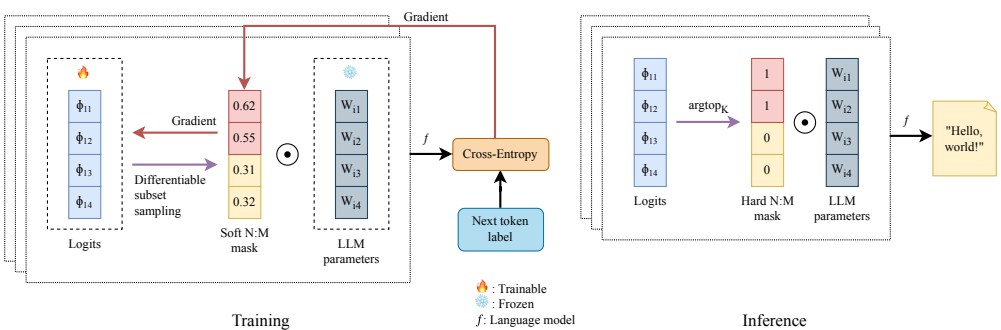

Figure 2: Overview of the SUSI Framework for Semi-Structured Pruning via Differentiable Subset Sampling, illustrating the training and inference phases.

However, such an optimization problem is NP-hard due to the vast search space, where there exists $\binom{M}{N}^G$ feasible solutions. In the context of Large Language Models, the number of weight groups $G$ is gargantuan, making this combinatorial optimization problem impractical to brute-force. Therefore, in the following section, we reformulate the above problem as a stochastic variational optimization variant to gain tractability and improve efficiency.

### 3.2 SUSI: Semi-Structured Pruning via Differentiable Subset Sampling

The overview of SUSI is illustrated in Figure 2. Accordingly, stochastic variational optimization (Bird et al., 2018) is based on an observation that given an arbitrary distribution $q(x)$ the expectation of a function $f(x)$ provides an upper bound on its minimum:

$$\min_x f(x) \leq \mathbb{E}_{q(x)}[f(x)] \tag{6}$$

By treating pruning masks as random variables, the optimization problem in the Equation 5 can be reframed as minimizing the variational upper bound of the objective with respect to the variational distribution parameters. Formally, we seek to find:

$$\mathbf{\Phi}^* = \underset{\mathbf{\Phi}}{\operatorname{argmin}} \, \mathbb{E}_{P(\mathbf{M}|\mathbf{\Phi})}[\mathcal{L}_{\mathrm{CE}}(\mathcal{D}; \mathbf{W} \odot \mathbf{M})] \tag{7}$$

where $\mathbf{\Phi} = \{\phi_1, \ldots, \phi_G\}$ is a set of parameters, corresponding to variational distributions $P(\mathbf{m}_1|\phi_1), \ldots, P(\mathbf{m}_G|\phi_G)$, with joint distribution $P(\mathbf{M}|\mathbf{\Phi}) = \prod_{i=1}^{G} P(\mathbf{m}_i|\phi_i)$. Through this formulation as a stochastic variational optimization problem, the sampling of masks can be reparameterized and relaxed to be a differentiable function with respect to variational distributions' parameters, making it possible to learn via gradient-based optimization.

#### 3.2.1 Variational Distribution Selection

Since pruning masks are $N$-hot vectors of length $M$, each mask can take one of $\binom{M}{N}$ possible values. Modeling such a distribution over possible values requires $\binom{M}{N} - 1$ free parameters, which grows combinatorially as $M$ increases. To efficiently learn masks with a reasonable number of parameters, we propose using the WRS distribution (Equation 1) over ordered subsets to model mask distributions $P(\mathbf{m}_i|\phi_i)$. Let $\mathcal{S}_i = \{\boldsymbol{\mu}_i^{(1)}, \ldots, \boldsymbol{\mu}_i^{(N)}\}$ be a set of $N$ one-hot vectors representing selected weights within the $i$-th group, sampled from the WRS distribution using the Gumbel-Top-$K$ trick, the probability of a mask $\mathbf{m}_i$ is then:

$$P(\mathbf{m}_i|\phi_i) = \sum_{\mathcal{S}_{\mathbf{m}_i}} P_{\mathrm{WRS}}(\mathcal{S}_{\mathbf{m}_i}|\exp(\phi_i)) \tag{8}$$

where $\mathcal{S}_{\mathbf{m}_i}$ denotes the set of elements whose sum equals $\mathbf{m}$, where the $N$-hot mask $\mathbf{m}_i$ can be obtained by summing up $\boldsymbol{\mu}_i^{(j)}$s. Exactly computing this probability is expensive and unnecessary since constructing $\mathbf{m}_i$ ignores the order of items in the sampled subset, and the expected loss can

be computed via Monte Carlo sampling. The original problem then turns into learning to select important weights, with importance score $\exp(\phi_{ij})$, in order to minimize the objective. Denoting the WRS distribution of an arbitrary subset $\mathcal{S}_i$ as $P_{\text{WRS}}(\mathcal{S}_i|\phi_i)$ for brevity, the optimization problem (Equation 7) is then reformulated as follows:

$$\mathbf{\Phi}^* = \underset{\mathbf{\Phi}}{\arg\min} \; \mathbb{E}_{P_{\text{WRS}}(\mathcal{S}|\mathbf{\Phi})}[\mathcal{L}_{\text{CE}}(\mathcal{D}; \mathbf{W} \odot \mathbf{M})] \tag{9}$$

where $\mathcal{S} = \{\mathcal{S}_1, \ldots, \mathcal{S}_G\}$ is a collection of $G$ subsets, generated from the joint distribution $P_{\text{WRS}}(\mathcal{S}|\mathbf{\Phi}) = \prod_{i=1}^{G} P_{\text{WRS}}(\mathcal{S}_i|\phi_i)$. Each $N$-hot pruning mask $\mathbf{m}_i$ in the collection $\mathbf{M}$ is constructed as $\mathbf{m}_i = \sum_{\boldsymbol{\mu}_i^{(j)} \in \mathcal{S}_i} \boldsymbol{\mu}_i^{(j)}$. Parameterizing masks as sums of subsets sampled from WRS-restricted distributions yields the same expected loss as sampling masks from the exact distributions, as proved in Theorem 1. Our approach reduces the parameter complexity by reformulating the $N$-hot mask sampling process as a sequential sampling without replacement paradigm. Instead of maintaining a full categorical distribution over $\binom{M}{N}$ configurations, we model only a single categorical distribution over every $M$ model parameters, requiring exactly $M$ parameters regardless of $N$. The proposed method achieves a reduction in parameter complexity from $\mathcal{O}\left(\binom{M}{N}\right)$ to $\mathcal{O}(M)$, representing an exponential improvement in memory efficiency.

### 3.2.2   MASK SELECTION RELAXATION

To make the objective differentiable with respect to variational distributions' parameters, we relax the sampling process using the Gumbel-Top-$K$ trick. Given logits $\phi_i = [\phi_{i1}, \ldots, \phi_{iM}]$ forming a categorical distribution over $M$ consecutive model weights within the $i$-th group, the probability of selecting the $j$-th weight is achieved via $\text{softmax}$: $\pi_{ij} = \exp(\phi_{ij})/\sum_{k=1}^{M} \exp(\phi_{ik})$. To sample a subset $\mathcal{S}_i$ without replacement from this distribution, we first perturb logits with Gumbel noise independently to attain random keys:

$$\kappa_{ij} = \phi_{ij} + g_{ij}, \quad g_{ij} \overset{\text{i.i.d}}{\sim} \text{Gumbel}(0,1) \tag{10}$$

We define the adjusted keys of the $i$-th group at sampling step $k$ as $\boldsymbol{\alpha}_i^{(k)} = [\alpha_{i1}^{(k)}, \ldots, \alpha_{iM}^{(k)}]$, The update rule follows the Gumbel-Top-$K$ procedure, except that we incorporate a power term $p > 1$ to amplify the impact of removing the selected item in the previous sampling step. This modification improves stability during training. Formally:

$$\boldsymbol{\alpha}_i^{(1)} := [\kappa_{i1}, \ldots, \kappa_{iM}], \quad \boldsymbol{\alpha}_i^{(k)} := \boldsymbol{\alpha}_i^{(k-1)} - |\log(1 - \boldsymbol{\mu}_i^{(k-1)})|^p \tag{11}$$

Finally, an approximated relaxed one-hot vector $\boldsymbol{\mu}_i^{(k)} = [\mu_{i1}^{(k)}, \ldots, \mu_{iM}^{(k)}]$ representing the selected item at the $k$-th sampling step is achieved by taking $\text{softmax}$ over adjusted keys with temperature $\tau$:

$$\mu_{ij}^{(k)} = \frac{\exp(\alpha_{ij}^{(k)}/\tau)}{\sum_{k=1}^{M} \exp(\alpha_{ik}^{(k)}/\tau)} \tag{12}$$

After $N$ sampling steps, a set of soft one-hot vectors representing selected weights is attained. By summing up these vectors, a relaxation of the $N$-hot pruning mask can be constructed, enabling gradient-based training.

### 3.2.3   TEMPERATURE ANNEALING

The temperature $\tau$ is mentioned as a hyperparameter controlling the hardness of one-hot approximations. Additionally, we define a hyperparameter $\lambda$, which regulates the degree of randomness in the sampling process. Subsequently, the Gumbel-Top-$K$ trick is applied to the scaled logits, denoted as $\overline{\phi}_i = \phi_i/\lambda$. In our experiments, we implement an annealing schedule for $\tau$ and $\lambda$ to guide the mask learning process, beginning with high randomness to promote solution exploration and converging to a small set of optimal solutions by training's end. We adopt a linear annealing schedule, where at the $t$-th training step, the temperatures are defined as follows:

$$\tau_t = \tau_{\text{init}} \times (1 - \frac{t}{T}) + \tau_{\text{end}} \times \frac{t}{T}; \quad \lambda_t = \lambda_{\text{init}} \times (1 - \frac{t}{T}) + \lambda_{\text{end}} \times \frac{t}{T} \tag{13}$$

Table 1: Comparative evaluation of zero-shot accuracy across multiple benchmark datasets for various pruning methods applied to OPT models of different sizes with 2:4 sparsity pattern. Bold values denote the highest performance in each metric. The column 'W/U' indicates whether weight updates are applied during pruning.

| Method | W/U | ARC-C | ARC-E | HellaS. | PIQA | RACE | SciQ | Average |
|---|---|---|---|---|---|---|---|---|
| **Base Model: OPT-125M** | - | 19.03 | 43.52 | 29.19 | 62.95 | 30.05 | 75.20 | 43.32 |
| Magnitude | ✗ | 17.66 | 32.28 | 27.14 | 57.67 | 22.78 | 44.00 | 33.59 |
| Wanda (Sun et al., 2024) | ✗ | 18.69 | 36.03 | 27.55 | 59.09 | 23.54 | 64.70 | 38.27 |
| SparseGPT (Frantar & Alistarh, 2023) | ✓ | **19.71** | 38.09 | **27.60** | 59.74 | 25.55 | 69.00 | 39.95 |
| MaskLLM (Fang et al., 2024) | ✗ | 18.34 | 39.73 | 27.50 | 61.26 | 26.79 | **70.30** | 40.65 |
| **SUSI** (Ours) | ✗ | 19.02 | **40.19** | 27.33 | 61.97 | **28.04** | 69.80 | **41.06** |
| **Base Model: OPT-350M** | - | 20.82 | 44.02 | 32.02 | 64.58 | 29.95 | 74.90 | 44.38 |
| Magnitude | ✗ | 16.72 | 31.52 | 27.09 | 57.40 | 22.87 | 51.30 | 34.48 |
| Wanda (Sun et al., 2024) | ✗ | **19.71** | 34.64 | **28.76** | 60.34 | 26.79 | 64.70 | 39.16 |
| SparseGPT (Frantar & Alistarh, 2023) | ✓ | 18.52 | 34.89 | 28.43 | 59.58 | 26.89 | **66.60** | 39.15 |
| MaskLLM (Fang et al., 2024) | ✗ | 18.26 | 37.71 | 27.52 | **61.48** | 26.99 | **66.60** | 39.76 |
| **SUSI** (Ours) | ✗ | 18.17 | **38.42** | 27.85 | 61.15 | **27.85** | 66.20 | **39.94** |
| **Base Model: OPT-1.3B** | - | 23.29 | 57.03 | 41.54 | 71.76 | 34.16 | 84.30 | 52.01 |
| Magnitude | ✗ | 17.83 | 39.31 | 31.15 | 61.81 | 26.22 | 65.40 | 40.29 |
| Wanda (Sun et al., 2024) | ✗ | 20.82 | 47.60 | 33.85 | 65.72 | 30.53 | **79.40** | 46.32 |
| SparseGPT (Frantar & Alistarh, 2023) | ✓ | **21.93** | 45.62 | **34.19** | 63.98 | 32.06 | 78.90 | 46.11 |
| MaskLLM (Fang et al., 2024) | ✗ | 19.53 | 47.39 | 33.29 | **66.76** | 31.87 | 76.40 | 45.87 |
| **SUSI** (Ours) | ✗ | 21.67 | **47.68** | 33.50 | 66.70 | **32.15** | 77.20 | **46.48** |

## 4 EXPERIMENT

### 4.1 EXPERIMENTAL SETTING

The proposed method is evaluated on three OPT models (Zhang et al., 2022) of increasing sizes (e.g., 125M, 350M, and 1.3B parameters) to assess its stability and scalability under semi-structured pruning. All main experiments adopt a 2:4 sparsity pattern, compatible with NVIDIA Ampere hardware. The detailed hyperparameters are listed in the Appendix A.3. Training runs for 2,000 steps with a batch size of 256 and a sequence length of 2048, processing 1B tokens in total. To ensure robust generalization, training data is collected on 1B tokens sampled from the C4 corpus (Raffel et al., 2020), a cleaned English dataset aligned with OPT's pretraining data.

To assess the effectiveness of the proposed approach, four representative semi-structured pruning methods are selected, covering a range of popular strategies from classical to recent advancements: i) **Magnitude** is a simple, data-free method that removes parameters with the smallest absolute values. While this method is easy to implement, it often yields subpar results due to the limitations of parameter sensitivity and model dynamics; ii) **Wanda** (Sun et al., 2024) combines parameter magnitudes with activation statistics at each layer, achieving better performance than pure magnitude pruning, especially at higher sparsity, while maintaining computational efficiency; iii) **SparseGPT** (Frantar & Alistarh, 2023) incorporates activation outputs and Hessian information to estimate parameter importance, followed by parameter updates to reduce output error further. This method yields high accuracy but is more computationally demanding; and iv) **MaskLLM** (Fang et al., 2024) is quite similar to the proposed method in this study, by learning pruning masks with minimizing calibration loss under an N:M sparsity constraint, modeled via a multinomial distribution. It delivers strong performance across benchmarks but suffers from high computational cost.

### 4.2 MAIN RESULTS

Table 1 presents zero-shot accuracy results across various benchmark datasets. SUSI consistently achieves the highest or near-highest average accuracy across all evaluated OPT models (41.06% for OPT-125M, 39.94% for OPT-350M, and 46.48% for OPT-1.3B), outperforming baselines such as Magnitude, Wanda, SparseGPT, and MaskLLM. Notably, SUSI exhibits a minimal performance drop compared to unpruned models (e.g., 19.02% vs. 19.03% on ARC-C for OPT-125M), highlighting its ability to preserve model quality through differentiable subset sampling. This advantage over heuristic-based methods like Magnitude becomes more pronounced as model size increases (e.g.,

46.48% for SUSI vs. 45.87% for MaskLLM on OPT-1.3B), highlighting its scalability. Additionally, SUSI shows strong performance across diverse tasks (e.g., 27.85% on RACE for OPT-350M), effectively balancing sparsity and accuracy while maintaining lower computational overhead compared to MaskLLM. Table 2 reports the PPL performance on WikiText-2, highlighting the advantages of the proposed method as follows:

**(i) Effectiveness of Differentiable Subset Sampling:** SUSI consistently outperforms other baselines across all model scales, suggesting that the proposed differentiable subset sampling mechanism effectively learns performant sparsity patterns, better preserving model quality post-pruning.

Furthermore, the small gap between the perplexity of the pruned SUSI models and the unpruned baseline indicates that SUSI maintains competitive performance even under aggressive pruning settings.

**(ii) Scalability across Model Sizes:** SUSI demonstrates consistent improvements across increasing model scales, showing especially strong results for medium (OPT-350M) and large (OPT-1.3B) models. This indicates that the method generalizes well and maintains scalability, which is often a limitation of recent pruning methods.

Table 2: Perplexity scores on WikiText-2.

| Method | OPT-125M | OPT-350M | OPT-1.3B |
|---|---|---|---|
| **w/o Pruning** | 31.95 | 25.42 | 16.41 |
| Magnitude | 407.66 | 655.87 | 245.75 |
| Wanda | 92.50 | 134.26 | 34.09 |
| SparseGPT | 72.80 | 61.23 | 29.27 |
| MaskLLM | 50.91 | 55.86 | 28.56 |
| **SUSI** (Ours) | **50.24** | **54.14** | **28.05** |

**(iii) Robustness of Learnable Mask Approaches:** While Traditional magnitude pruning performs poorly (e.g., 655.87 PPL on OPT-350M), reaffirming that naive pruning strategies significantly degrade language modeling performance. The promising performance of SUSI and MaskLLM emphasizes the importance of structured and learnable pruning mechanisms.

## 4.3 DETAILED ANALYSIS

### 4.3.1 EFFICIENT TRAINING

Figure 3 presents a comparative analysis of the parameter efficiency and data efficiency achieved by the proposed SUSI method under both 2:4 and 2:8 sparsity settings. Figure 3(a) reports the number of trainable parameters across multiple OPT model sizes. Under the 2:4 pattern, SUSI consistently requires about $1.5\times$ fewer parameters than MaskLLM, effectively lowering optimization costs. More importantly, the advantage of SUSI becomes even more evident in the 2:8 setting: while MaskLLM requires up to $4.2$B parameters for OPT-1.3B, SUSI reduces this to $1.2$B, achieving a $3.5\times$ reduction. Such parameter efficiency directly translates into substantial computational and memory savings, which is critical for deployment in resource-constrained environments. Sequentially, Figure 3(b) reports the perplexity on WikiText-2 as a function of the number of training tokens for the OPT-350M model. Under the 2:4 pattern, SUSI consistently achieves lower perplexity than MaskLLM across all token budgets, demonstrating superior data efficiency. In the 2:8 pattern, although the number of trainable parameters is drastically reduced compared to 2:4 (Figure 3a), SUSI maintains competitive perplexity with MaskLLM, reaching $144.9$ at 1B tokens. These results highlight the robustness of SUSI: it not only improves parameter efficiency but also sustains competitive modeling performance under more aggressive sparsity constraints. The detailed performance of 2:8 sparsity patterns is shown in the Appendix A.7.

### 4.3.2 ABLATION STUDY

Figure 4 summarizes the ablation study on the proposed SUSI model, focusing on the contributions of two critical design choices: (i) the power term $p$, which amplifies the effect of removing a selected weight (Equation 11); and (ii) the temperature annealing schedule that gradually sharpens the sampling distribution (Equation 13). Figure 4(a) illustrates the training loss trajectories under different configurations. Without the power term ($p = 1.0$), convergence is noticeably slower and less stable, with higher final loss compared to $p = 3.0$. Increasing $p$ strengthens the penalization on selected weights, which accelerates convergence and consistently lowers the final loss, suggesting that this

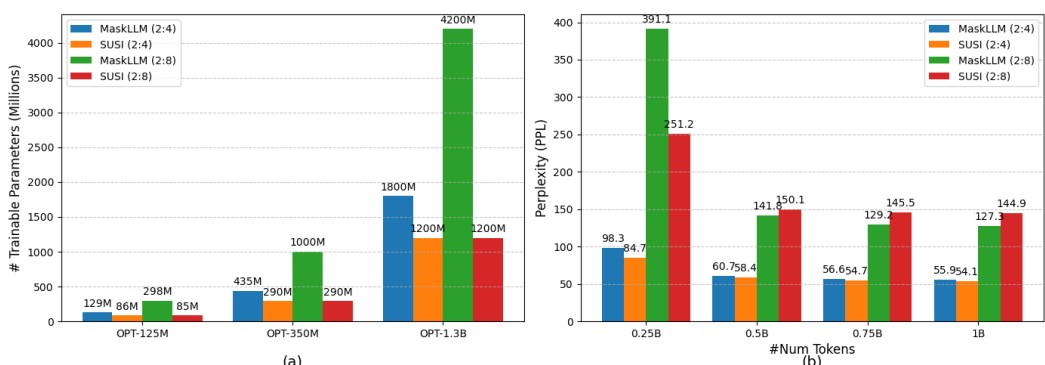

(a)                                                                      (b)

Figure 3: Comparison of sparsity and perplexity performance: (a) Learnable parameter counts under the 2:4 and 2:8 sparsity settings across multiple OPT model sizes; and (b) Perplexity versus number of training tokens on Wikitext-2 for the OPT-350M model.

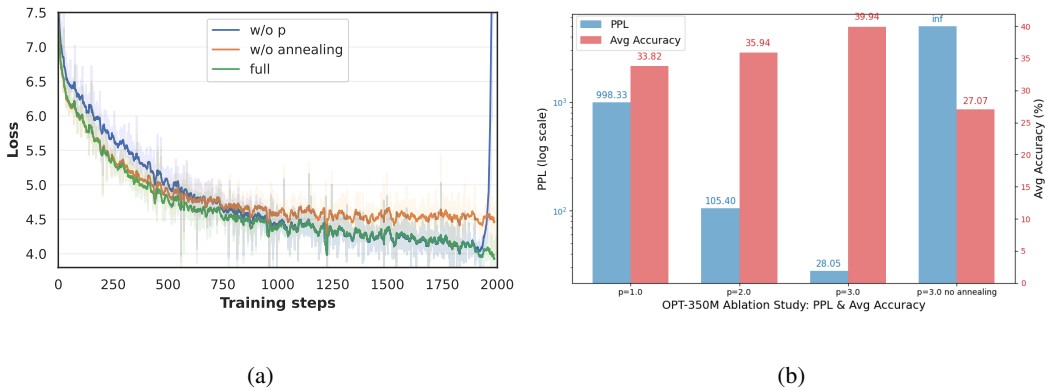

(a)                                                                      (b)

Figure 4: Comparison of training dynamics and ablation results: (a) shows the training loss convergence across different configurations (with/without $p$ and annealing). (b) ablation study on OPT-350M showing PPL (log-scale) and average accuracy.

mechanism facilitates escaping suboptimal mask distributions. On the other hand, removing the annealing mechanism leads to rapid divergence, underscoring the necessity of temperature scheduling for maintaining a stable optimization process. Figure 4(b) reports the downstream performance on OPT-350M in terms of perplexity (log scale) and average zero-shot accuracy. As $p$ increases from 1.0 to 3.0, perplexity drops dramatically (from 998.33 to 28.05) and accuracy improves significantly (from 33.82% to 39.94%), validating the importance of the power term for effective mask learning. In contrast, disabling annealing results in infinite perplexity and a severe accuracy drop (27.07%), highlighting that annealing is indispensable for stable training and generalization. The results demonstrate that both components are synergistic: the power term enhances selection sharpness, while annealing ensures convergence stability, which improves the performance.

### 4.3.3 ROBUSTNESS ANALYSIS

To further evaluate the stability and robustness of SUSI, we trained the variational mask parameters using three distinct random seeds (42, 123, 1812) and measured the overlap of learned pruning masks across key layers. Figure 5 reports the probability of mask overlap between runs for representative modules such as self_attn.q_proj, self_attn.k_proj, and mlp.up_proj. Accordingly, the learned pruning masks show high overlap across seeds (e.g., 0.88 for q_proj, 0.83 for k_proj, 0.94 for mlp.up_proj), and downstream performance varies by less than 0.5%. These results confirm that SUSI consistently converges to similar sparsity patterns with minimal variation across initializations, demonstrating strong robustness and reproducibility.

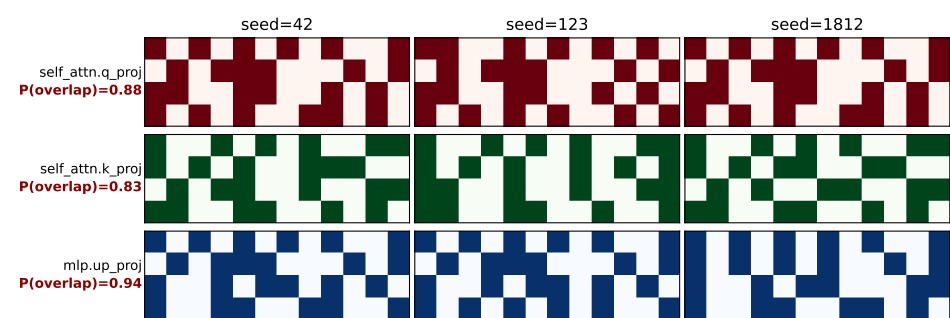

Figure 5: The learned masks from the query projection, key projection, and MLP up-projection in the first transformer block exhibit high similarity across different random seeds.

### 4.4 RELATED WORKS

Pruning LLMs is a critical optimization technique that removes less significant or redundant parameters, such as weights or neurons, from the neural network architecture. This process reduces model size, computational complexity, and memory requirements, thereby improving inference speed and enabling deployment on resource-constrained devices. Pruning methods for LLMs are broadly categorized into structured, unstructured, and semi-structured approaches, each with distinct characteristics and trade-offs (Cheng et al., 2024).

**Structured pruning** involves the elimination of entire architectural components, such as layers or attention heads, to improve computational efficiency (Ashkboos et al., 2024; Xia et al., 2024; An et al., 2024; Liu et al., 2025a; Le et al., 2025). This approach simplifies the model structure, making it more amenable to hardware optimization. However, it frequently results in substantial performance degradation, necessitating extensive retraining to restore model functionality.

**Unstructured pruning** targets individual weights based on their significance, enabling high performance even at elevated sparsity levels (Dong et al., 2024; Sun et al., 2024). Despite its efficacy in preserving model accuracy, the irregular sparsity patterns produced are often incompatible with hardware acceleration, limiting its practical applicability in deployment scenarios.

**Semi-structured pruning** has emerged as a promising approach, striking a balance between the benefits of structured and unstructured methods. By enforcing regular sparsity patterns, such as $N{:}M$ sparsity, this technique optimizes models for hardware acceleration while maintaining performance (Hubara et al., 2021). Methods like SparseGPT (Frantar & Alistarh, 2023) and Wanda (Sun et al., 2024) employ training-free pruning, achieving efficiency without retraining. More recent methods, such as MaskLLM (Fang et al., 2024) and AST (Huang et al., 2025), focus on retraining sparse LLMs, which achieve promising performances while maintaining hardware compatibility. Nonetheless, the significant computational overhead associated with the number of trainable parameters remains a critical challenge, warranting further investigation. Building on this foundation, our proposed method leverages weighted reservoir sampling to enhance semi-structured pruning with $N{:}M$ sparsity, aiming to enable the retraining of semi-structured sparse LLM with minimal training costs.

## 5 CONCLUSION

This study introduced SUSI, a novel semi-structured pruning technique for LLMs, utilizing differentiable subset sampling to efficiently derive N:M sparsity masks. Compared to existing methods, SUSI reduces the number of trainable parameters and associated memory overhead while maintaining strong performance. Experiments on OPT models (125M, 350M, 1.3B parameters) show that SUSI outperforms existing methods in perplexity on the Wikitext-2 dataset and maintains competitive zero-shot accuracy across a range of benchmarks. Additionally, SUSI exhibits enhanced data efficiency and scalability as calibration data increases. These results establish SUSI as a promising solution for compressing LLMs, effectively balancing performance retention with the demands of resource-constrained deployment environments.

## REPRODUCIBILITY STATEMENT

We have taken several steps to ensure the reproducibility of our work:

**Datasets**. All training and calibration data used in our experiments are publicly available. We follow prior work by sampling 1B tokens from the cleaned English portion of the C4 corpus for calibration and training, ensuring alignment with OPT's pretraining distribution. For evaluation, we employ well-known open-source benchmarks, including WikiText-2 for perplexity evaluation and ARC (Easy/Challenge), HellaSwag, PIQA, SciQ, and RACE for zero-shot task accuracy. Dataset statistics and details are provided in Appendix A.1 to facilitate replication.

**Code and Implementation**. We provide an anonymous, fully reproducible implementation of SUSI, including (i) training scripts for variational mask optimization, (ii) hyperparameter configurations (see Appendix A.2), and (iii) evaluation scripts leveraging the LM-Evaluation-Harness toolkit. All results reported in this paper can be reproduced using the provided codebase.

**Availability**. To encourage transparency and facilitate verification of our findings, we submit the source code and experiment configuration files as supplementary material. An anonymous and reproducible version of the repository can be accessed at the following link: `https://anonymous.4open.science/r/susi-2E2C`.

This repository contains all necessary scripts, instructions, and environment configuration files (including requirements.txt) for reproducing our results end-to-end on standard hardware.

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

## A  APPENDIX

### A.1  WRS YIELDS EQUIVALENT VARIATIONAL OBJECTIVE

**Theorem 1.** *Let $P(\mathbf{m}_i|\boldsymbol{\phi}_i)$ be the exact distribution of each mask $\mathbf{m}_i$ defined as in Equation 8. The expected loss when sampling each mask from its exact distribution is equivalent to the expected loss obtained when each mask is parameterized as a sum of elements in an ordered subset $\mathcal{S}_i$ sampled from the corresponding restricted distribution $P_{WRS}(\mathcal{S}_i|\boldsymbol{\phi}_i)$.*

*Proof.* Without loss of generality, we prove the following terms are equivalent:

$$\mathbb{E}_{P(\mathbf{m}|\boldsymbol{\phi})}[f(\mathbf{m})] = \mathbb{E}_{P_{\text{WRS}}(\mathcal{S}|\boldsymbol{\phi})}\left[f\left(\sum_{\boldsymbol{\mu}\in\mathcal{S}}\boldsymbol{\mu}\right)\right] \tag{14}$$

where $f(\mathbf{m})$ is an objective function depending on $\mathbf{m}$, $P(\mathbf{m}|\boldsymbol{\phi}) = \sum_{\mathcal{S}_{\mathbf{m}}} P_{\text{WRS}}(\mathcal{S}_{\mathbf{m}}|\boldsymbol{\phi})$ is the exact distribution with $\mathcal{S}_{\mathbf{m}}$s are sets that the sum of elements in $\mathcal{S}_{\mathbf{m}}$ equals $\mathbf{m}$.

Given $\mathcal{M}$, the set of binary masks satisfying the N:M sparsity, the expected loss when sampling $\mathbf{m}$ from the exact distribution is then:

$$
\begin{aligned}
\mathbb{E}_{P(\mathbf{m}|\phi)}[f(\mathbf{m})] &= \sum_{\mathbf{m}\in\mathcal{M}} P(\mathbf{m}|\phi)f(\mathbf{m}) = \sum_{\mathbf{m}\in\mathcal{M}} \left(\sum_{\mathcal{S}_{\mathbf{m}}} P_{\text{WRS}}(\mathcal{S}_{\mathbf{m}}|\phi)\right) f\left(\sum_{\boldsymbol{\mu}\in\mathcal{S}_{\mathbf{m}}} \boldsymbol{\mu}\right) \\
&= \sum_{\mathbf{m}\in\mathcal{M}} \sum_{\mathcal{S}_{\mathbf{m}}} P_{\text{WRS}}(\mathcal{S}_{\mathbf{m}}|\phi)f\left(\sum_{\boldsymbol{\mu}\in\mathcal{S}_{\mathbf{m}}} \boldsymbol{\mu}\right) = \sum_{\mathbf{m}\in\mathcal{M}} P_{\text{WRS}}(\mathcal{S}|\phi)f\left(\sum_{\boldsymbol{\mu}\in\mathcal{S}} \boldsymbol{\mu}\right) \quad (15) \\
&= \mathbb{E}_{P_{\text{WRS}}(\mathcal{S}|\phi)}\left[f\left(\sum_{\boldsymbol{\mu}\in\mathcal{S}} \boldsymbol{\mu}\right)\right]
\end{aligned}
$$

$\square$

The final expression is precisely the expectation of $f$ under the distribution $P_{\text{WRS}}(\mathcal{S}|\phi)$, proving the claim.

## A.2 EVALUATION METRICS AND BENCHMARK DATASETS

Following previous works in this research field, three automated metrics are considered for the evaluation, including both quantitative and qualitative metrics to capture the full impact of pruning: i) *Task Accuracy (ACC)*: on common NLP tasks such as question answering in reading comprehension, mathematics, and science. These tasks are typically assessed in zero-shot or few-shot settings using benchmark datasets; *Perplexity (PPL)*: is a standard metric for assessing language model quality. It

Table 3: Statistics of datasets used for zero-shot evaluation.

| Dataset | Questions | Task Type |
|---|---|---|
| ARC-Easy | 2,376 | Multiple-choice science |
| ARC-Challenge | 1,172 | Multiple-choice science |
| HellaSwag | 10,042 | Sentence completion |
| PIQA | 1,838 | Physical interaction QA |
| RACE | 1,045 | Multiple-choice comprehension |
| SciQ | 1,000 | Multiple-choice science |

measures how well the model predicts the next word in a sequence, with lower values indicating better predictive performance. The benchmark datasets used to assess the effectiveness of pruning methods include WikiText-2 (Merity et al., 2017) for perplexity evaluation and a range of NLP benchmark datasets for zero-shot evaluation, which cover diverse task types and reasoning requirements, including ARC (Clark et al., 2018), HellaSwag (Zellers et al., 2019), PIQA (Bisk et al., 2020), SciQ (Welbl et al., 2017), and RACE (Lai et al., 2017). These evaluations are conducted using the LM-Evaluation-Harness toolkit (Gao et al., 2024).

Table 3 provides a comprehensive summary of the datasets used for zero-shot evaluation across multiple tasks. These datasets span a range of domains, including commonsense reasoning, science question answering, and reading comprehension, thereby ensuring a rigorous and diverse assessment of pruning performance.

## A.3 HYPERPARAMETER SETTING

The hyperparameters used for training SUSI are listed in Table 4. These settings were carefully chosen to balance convergence stability and computational efficiency across all evaluated models. Specifically, model weights remain frozen during training. The variational distribution is initialized from a standard normal ($\mu = 0.0$, $\sigma = 0.01$), and a simulated annealing process gradually reduces randomness. Temperatures $\tau$ and $\lambda$ linearly decay from 1.0 to 0.05 and from 1.0 to 0.002, respectively. Optimization uses AdamW-8bit with a learning rate decaying from $1 \times 10^{-3}$ to $1 \times 10^{-4}$,

Table 4: Hyperparameter configuration used in training.

| Parameter | Values |
|---|---|
| Initialization distribution | $\mathcal{N}(0, 0.01)$ |
| Gumbel-Softmax temperature | $\tau = 1.0 \rightarrow 0.05$ |
| Sampling temperature | $\lambda = 1.0 \rightarrow 0.002$ |
| Weight decay | 0.05 |
| Learning rate | $10^{-3} \rightarrow 10^{-4}$ |
| Strengthening power term | $p = 3.0$ |
| AdamW parameters | $\beta_1 = 0.9, \beta_2 = 0.95$ |
| Batch size | 256 |
| Sequence length | 2048 |
| Training steps | 2000 |

weight decay of 0.05, and $\beta_1 = 0.9$, $\beta_2 = 0.95$, matching the OPT pretraining setup. The power term $p$ (Equation 11) is selected from $\{1, 2, 3\}$, where $p = 1$ corresponds to no power-term scaling, and larger values of $p$ (e.g., $p = 2$ or 3) progressively emphasize the impact of removing higher-importance elements.

## A.4 GUMBEL-TOP-K ALGORITHM

The algorithm for Gumbel-Top-K is illustrated in the Algorithm 1. Specifically, we provide a clear description of the Gumbel-Top-K sampling procedure employed to enable differentiable mask learning. This formulation allows for efficient sampling of K items without replacement while preserving differentiability for gradient-based optimization.

---

**Algorithm 1** Gumbel-Top-$K$ Sampling Algorithm (Differentiable)

---

**Input**: Set of candidates $\mathcal{X} = \{x_1, \ldots, x_N\}$ with corresponding logits $\boldsymbol{\phi} = [\phi_1, \ldots, \phi_N]$, number of samples $K$, temperature $\tau > 0$

**Output**: Soft $K$-hot selection vector $\mathbf{S} \in \mathbb{R}^N$

1: **for** $i \leftarrow 1$ to $N$ **do**
2:    $u_i \sim \text{Uniform}(0, 1)$
3:    $g_i \leftarrow -\log(-\log(u_i))$                *// Sample Gumbel noise*
4:    $\kappa_i \leftarrow \phi_i + g_i$                    *// Compute perturbed key*
5: **end for**
6: $\boldsymbol{\alpha}^{(1)} \leftarrow [\kappa_1, \ldots, \kappa_N]$
7: **for** $k \leftarrow 1$ to $K$ **do**
8:    $\boldsymbol{\mu}^{(k)} \leftarrow \text{softmax}(\boldsymbol{\alpha}^{(k)}/\tau)$
9:    $\boldsymbol{\alpha}^{(k+1)} \leftarrow \boldsymbol{\alpha}^{(k)} + \log(1 - \boldsymbol{\mu}^{(k)})$
10: **end for**
11: $\mathbf{S} \leftarrow \sum_{k=1}^{K} \boldsymbol{\mu}^{(k)}$                *// Soft $K$-hot vector*
12: **return S** =0

---

## A.5 COMPARISON TO STRAIGHT-THROUGH GUMBEL-TOP-$K$

We further examined whether adopting a straight-through (ST) Gumbel-Top-K estimator benefits pruning performance. In this variant, the forward pass generates discrete masks by directly applying an argtopK over Gumbel-perturbed logits, while the backward pass propagates gradients through the continuous Gumbel-softmax relaxation. This strategy enforces discretization earlier in training, which is able to improve mask interpretability. However, our empirical results in Table 5 show that ST Gumbel-Top-K leads to slightly inferior performance compared to the pure soft relaxation. For instance, on OPT-125M, ST achieves 51.20 perplexity and 40.04% average accuracy, while the soft approach reaches 50.24 perplexity and 41.06% accuracy. Similarly, on OPT-350M, the gap widens (60.49 vs. 54.14 perplexity). These observations suggest that the bias introduced by the ST estimator hampers generalization, outweighing the potential benefits of earlier discretization. Overall, the soft

Table 5: Comparison of pruning results with 2:4 sparsity, both with and without ST Gumbel-Top-$K$ estimator (denoted as "w STE" and "w/o STE").

| | **OPT-125M** | | **OPT-350M** | |
| Metric | w STE | w/o STE (ours) | w STE | w/o STE (ours) |
|---|---|---|---|---|
| PPL ($\downarrow$) | 51.20 | **50.24** | 60.49 | **54.14** |
| Avg. Acc ($\uparrow$) | 40.04 | **41.06** | 39.11 | **39.94** |

Gumbel-Top-K relaxation used in SUSI provides a more effective balance between trainability and performance.

## A.6 MASK DIFFERENCE ANALYSIS

To investigate how different pruning strategies select weights, we measure the overlap between masks produced by various methods on the same model. Figure 6 shows that SUSI's learned masks achieve much higher cross-seed similarity (82%) compared to one-shot pruning methods such as Magnitude (63%), Wanda (66%), and SparseGPT (75%), which produce substantially different sparsity patterns.

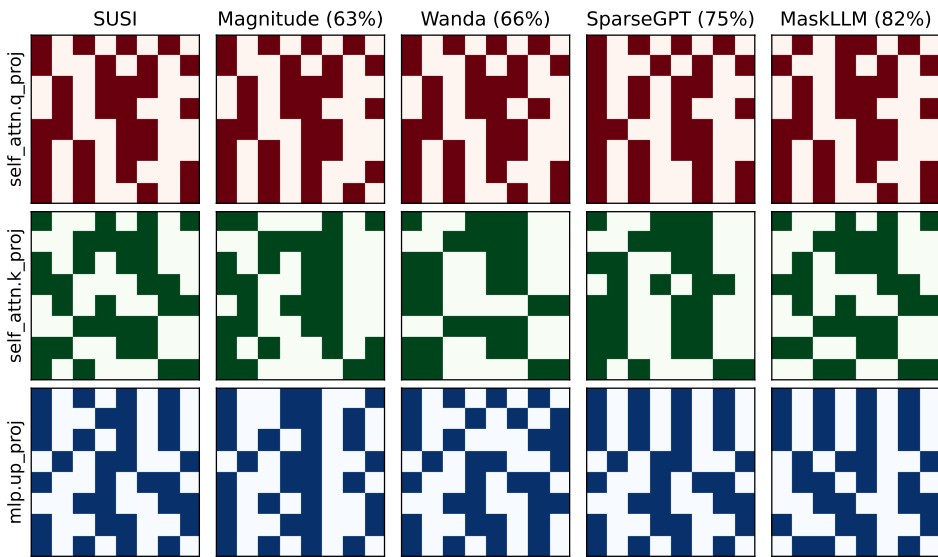

Figure 6: Mask difference analysis between SUSI and previous works. Besides the name of each baseline, place an overlapping percentage indicating the similarity of the produced masks between that baseline and SUSI.

Interestingly, the mask similarity of SUSI closely matches that of other mask-learning approaches like MaskLLM, suggesting that iterative mask optimization converges toward a stable and consistent subset of important weights. Combined with the main results, higher mask similarity is correlated with better perplexity and zero-shot accuracy, underscoring that stable mask learning plays a key role in achieving superior downstream performance.

## A.7 EXTEND TO OTHER SPARSITY PATTERN

To further examine the generality of SUSI, we extend our evaluation beyond the commonly studied 2:4 configuration. These alternative settings introduce more aggressive pruning constraints and exacerbate the challenges faced by learnable mask methods such as MaskLLM, whose parameter overhead grows quadratically. In contrast, SUSI preserves linear complexity in $M$, enabling efficient scalability to larger group sizes.

Table 6: Performance on 2:8 sparsity pattern.

| Method | W/U | ARC-C | ARC-E | HellaS. | PIQA | RACE | SciQ | Average ↑ | PPL ↓ |
|---|---|---|---|---|---|---|---|---|---|
| **Base Model: OPT-125M** | - | 19.03 | 43.52 | 29.19 | 62.95 | 30.05 | 75.20 | 43.32 | 32 |
| Magnitude | ✗ | **21.25** | 27.26 | 25.90 | 53.65 | 21.82 | 21.80 | 28.61 | 13431 |
| Wanda | ✗ | 18.86 | 29.12 | 26.19 | 54.35 | 21.44 | 28.80 | 29.79 | 5195 |
| SparseGPT | ✓ | 19.88 | 28.28 | 26.43 | 55.01 | 23.73 | 32.40 | 30.96 | 986 |
| MaskLLM | ✗ | 18.77 | **35.19** | 26.86 | 58.16 | **23.44** | 61.20 | **37.27** | **107** |
| **SUSI** (Ours) | ✗ | 18.17 | 33.80 | **26.91** | **58.27** | **23.44** | **62.70** | 37.22 | 110 |
| **Base Model: OPT-350M** | - | 20.82 | 44.02 | 32.02 | 64.58 | 29.95 | 74.90 | 44.38 | 25.42 |
| Magnitude | ✗ | **19.97** | 28.07 | 26.31 | 53.54 | 22.20 | 30.00 | 30.02 | 9805 |
| Wanda | ✗ | 18.52 | 27.61 | 26.51 | 53.48 | 22.11 | 29.00 | 29.54 | 2956 |
| SparseGPT | ✓ | 17.49 | 28.83 | 26.50 | 54.30 | 23.44 | 37.00 | 31.26 | 1358 |
| MaskLLM | ✗ | 16.55 | **31.61** | 26.38 | **57.51** | **24.78** | **58.60** | **35.91** | **127** |
| **SUSI** (Ours) | ✗ | 16.30 | 29.50 | **26.61** | 57.02 | 24.69 | 57.20 | 35.22 | 145 |

Table 7: Performance on the 4:8 sparsity pattern. Note that experimenting on MaskLLM could not be executed on our infrastructure in this setting due to the excessive number of trainable parameters.

| Method | W/U | ARC-C | ARC-E | HellaS. | PIQA | RACE | SciQ | Average ↑ | PPL ↓ |
|---|---|---|---|---|---|---|---|---|---|
| **Base Model: OPT-125M** | - | 19.03 | 43.52 | 29.19 | 62.95 | 30.05 | 75.20 | 43.32 | 32 |
| Magnitude | ✗ | 18.09 | 34.72 | 27.55 | 58.32 | 23.25 | 57.4 | 36.56 | 205 |
| Wanda | ✗ | **19.11** | 37.42 | 27.74 | 59.85 | 26.22 | 67.10 | 39.57 | 61 |
| SparseGPT | ✓ | 18.77 | 39.06 | **27.94** | 61.15 | 27.75 | 71.10 | 40.96 | 54 |
| MaskLLM | ✗ | - | - | - | - | - | - | - | - |
| **SUSI** (Ours) | ✗ | **19.11** | **40.36** | 27.67 | **62.25** | **29.37** | **72.30** | **41.84** | **41** |
| **Base Model: OPT-350M** | - | 20.82 | 44.02 | 32.02 | 64.58 | 29.95 | 74.90 | 44.38 | 25 |
| Magnitude | ✗ | 16.81 | 33.12 | 27.74 | 58.32 | 22.78 | 56.90 | 35.95 | 221 |
| Wanda | ✗ | 17.83 | 35.86 | 28.81 | 60.83 | 25.17 | 66.30 | 39.13 | 71 |
| SparseGPT | ✓ | **18.52** | 36.57 | **29.56** | 61.32 | 27.85 | **69.10** | 40.49 | 46 |
| MaskLLM | ✗ | - | - | - | - | - | - | - | - |
| **SUSI** (Ours) | ✗ | 18.16 | **38.85** | 28.79 | **62.51** | **29.33** | 68.51 | **41.03** | **42** |

As shown in Figure 3 and Table 6, under the 2:8 sparsity pattern, SUSI achieves a $3.5\times$ reduction in trainable parameters relative to MaskLLM, while maintaining competitive perplexity. This demonstrates that even with substantially fewer learnable parameters than in the 2:4 case, SUSI continues to deliver robust language modeling performance. These results underscore the efficiency of differentiable subset sampling in handling larger sparsity patterns.

The 4:8 sparsity pattern (Table 7) presents an even more demanding setting. Here, MaskLLM fails to execute due to the prohibitive number of trainable parameters. By contrast, SUSI remains tractable, successfully completing training and yielding stable evaluation results. This highlights a distinct advantage of SUSI: its parameter efficiency not only improves training feasibility but also makes previously impractical sparsity patterns accessible to large-scale language models.

## A.8    EXTEND SUSI TO RECENT LLMS

We further extend SUSI to recent LLM architectures, including Qwen2.5-0.5B and Llama3.2-1B, to examine its generality beyond the OPT family. As shown in Table 8, SUSI remains feasible and efficient under these modern settings. While the performance gap relative to dense models is more pronounced than in the OPT series (e.g., Qwen2.5-0.5B drops from 55.33% accuracy at 22 PPL to 43.75% at 46 PPL after pruning), SUSI still achieves competitive results. Compared to the OPT family, where SUSI nearly matches the dense baseline, these results highlight that SUSI scales consistently to diverse architectures, maintaining tractable training and offering substantial efficiency gains even when accuracy trade-offs are larger in more recent models.

## A.9    LIMITATIONS

Despite the promising performance and efficiency demonstrated by SUSI, several limitations remain:

Table 8: Performance of SUSI on recent LLMs (Qwen2.5-0.5B and Llama3.2-1B). SUSI remains tractable, demonstrating scalability across architectures. Although the performance gap to dense models is larger than in the OPT family, SUSI preserves competitive accuracy with favorable perplexity-efficiency trade-offs.

| Method | ARC-C | ARC-E | HellaS. | PIQA | RACE | SciQ | Average ↑ | PPL ↓ |
|---|---|---|---|---|---|---|---|---|
| **Base Model:Qwen2.5-0.5B** | 29.18 | 64.48 | 40.53 | 70.35 | 34.64 | 92.80 | 55.33 | 22 |
| **SUSI** (Ours) | 18.34 | 45.54 | 30.56 | 64.15 | 28.52 | 75.40 | 43.75 | 46 |
| **Base Model:Llama3.2-1B** | 31.31 | 65.49 | 47.72 | 74.48 | 37.89 | 91.40 | 58.05 | 13 |
| **SUSI** (Ours) | 20.39 | 45.20 | 32.33 | 65.89 | 29.28 | 77.60 | 45.12 | 32 |
| **Base Model: OPT-1.3B** | 23.29 | 57.03 | 41.54 | 71.76 | 34.16 | 84.30 | 52.01 | 16 |
| **SUSI** (Ours) | 21.67 | 47.68 | 33.50 | 66.70 | 32.15 | 77.20 | 46.48 | 28 |

First, the deployment of semi-structured sparsity is inherently hardware-dependent. At present, substantial throughput gains are realized only on select platforms (e.g., AMD ROCm and certain NVIDIA Ampere and Hopper GPUs) where 2:4 structured sparsity is natively supported and accelerated at the kernel level. Although SUSI can, in principle, be extended to arbitrary $N:M$ sparsity patterns, its practical utility is constrained by the absence of hardware kernels and vendor-optimized libraries for ratios other than 2:4. On accelerators or CPUs lacking such specialized support, pruning yields only marginal reductions in memory footprint and fails to deliver meaningful inference speedup. This hardware dependency poses a significant challenge for widespread adoption in heterogeneous production environments, where deployment targets may vary.

Second, the current evaluation focuses exclusively on English-centric OPT models and a limited set of standard NLP benchmarks. Future research should investigate the applicability of SUSI to multilingual LLMs, larger-scale models, and domain-specific tasks (e.g., code generation, reasoning-intensive applications) to assess its generalization and scalability comprehensively.

