# OpenReview forum: "SUSI: Semi-Structured Pruning for LLMs via Differentiable Subset Sampling"
_ICLR.cc/2026/Conference — ICLR 2026 Conference Withdrawn Submission_

### Official Review · Reviewer_iYsi · 2025-10-23

**Soundness:** 2
**Presentation:** 3
**Contribution:** 2
**Rating:** 2
**Confidence:** 5

**Summary:**

This paper studies the problem of semi-structured pruning of LLMs. The authors propose to use a differentiable framework, unlike, for example, layer-wise approaches of some of the previous work. They propose to write the pruning problem as a stochastic optimization problem, which requires Monte Carlo sampling. To make the optimization problem differentiable, they use a Gumbel sampling trick. Additional discussions are provided regarding the smoothing temperature. Numerical experiments are presented for three models from the OPT family.

**Strengths:**

- I think the proposed method is reasonable overall.
- I think the paper's writing is good.

**Weaknesses:**

- Experiments: The paper significantly lack on the experimental section. Only OPT models are used which are outdated, and only models up to 1.3B parameters are studied. This is concerning regarding the computational efficiency of the method. The baselines studied are also outdated. Please compare with https://arxiv.org/abs/2406.07831 (available at https://github.com/linkedin/FMCHISEL) and https://arxiv.org/abs/2310.08915 (available at https://github.com/zyxxmu/DSnoT). Please also see my additional comments in the Questions section below. Given this paper is mostly experimental, this is sufficient for me to reject the paper, unless my concerns are addressed.

**Questions:**

- How can this approach be extended to other sparsity frameworks such as structured sparsity?
- A more detailed methodological comparison with MaskLLM will be appreciated.
- A more detailed ablation study of annealing (the effect of $\lambda,\tau$).
- Please explicitly report the runtime of your method compared to the baselines.
- Please include more modern models, with larger sizes in your experiments.
- How will baselines such as Wanda perform if the pruned models are retrained over 1B tokens, similar to the setting used for SUSI?

---

### Official Review · Reviewer_Lwdw · 2025-10-28

**Soundness:** 2
**Presentation:** 2
**Contribution:** 1
**Rating:** 2
**Confidence:** 4

**Summary:**

This work introduces SUSI, a differentiable method for semi-structured pruning that improves parameter efficiency and performance over MaskLLM; however, these claims are not fully supported by the empirical validation. The zero-shot results show inconsistent performance, often underperforming against baselines on specific tasks, which makes reliance on an average score an unconvincing measure of superiority. Furthermore, the pruning evaluation omits key contemporary baselines that may outperform SUSI, and the experimental scope is limited to older, smaller-scale OPT models (up to 1.3B), failing to demonstrate practical utility on larger, more relevant architectures. Consequently, the paper’s contribution would be significantly strengthened by more consistent results, comprehensive comparisons, and validation on modern, larger-scale models.

**Strengths:**

This work reformulates semi-structured pruning as a differentibal subset sampling problem. A key advantage over prior work like MaskLLM is its increased parameter efficiency, leading to reduced computational demands during training.

**Weaknesses:**

This work has several limitations that currently undermine its contributions. The practical motivation for exploring N:M sparsity patterns beyond the hardware-supported 2:4 is unclear, and the empirical validation is insufficient: the choice of benchmarks is questionable as base models fail to exceed random chance in ARC-Challenge, the model scale (up to 1.3B OPT) is too small to claim scalability, and key state-of-the-art baselines are missing from comparisons. Furthermore, the claims of "consistent" superiority and "robustness" are overstated, as they rely on aggregated averages over unstable per-task performance and limited stability tests.

**Questions:**

1. This work investigates N:M sparsity patterns, yet the practical motivation and hardware relevance for patterns beyond 2:4 (which has dedicated support on NVIDIA Ampere) remain unclear. Could you please clarify the rationale for exploring patterns like 2:8 or 1:4? Specifically, is the assumption that these offer a superior accuracy-efficiency trade-off despite lacking current hardware acceleration, or is the goal to propose patterns for future hardware to acieve more acceleration?

2. The use of the ARC-Challenge benchmark (1/4 choice) is questionable for this study, as the base OPT models (125M to 1.3B) fail to exceed the 25% random guess rate, indicating the task is too difficult to meaningfully evaluate the performance of these original and pruned models. This issue, combined with the exclusive use of the older OPT model family, significantly limits the validity and generalizability of the results. To strengthen the contribution, the evaluation should be expanded to include larger, more capable base models (such as LLaMA or Qwen) on a broader set of tasks where the unpruned models demonstrate substantive baseline performance.

3. In Lines 318-319, the claim that *"SUSI consistently achieves the highest or near-highest average accuracy"* appears overstated when examining the per-task results in Table 1. While SUSI achieves the highest aggregate average, its ranking is unstable across individual tasks, often not being the top performer. This suggests the overall average may be sensitive to the specific choice and difficulty weighting of the benchmark tasks rather than demonstrating true consistent superiority. To solidly support this claim, could you please either clarify the definition of "near-highest" with a more robust statistical analysis or strengthen the evidence by including a broader and more balanced set of evaluation tasks.

4. The evaluation of semi-structured pruning performance appears incomplete, as several important and highly relevant baselines for 2:4 sparsity (such as [1, 2, 3, 4]) are missing from the comparison. According to the results in Table 2, an initial comparison with the results reported in these works suggests that SUSI may not consistently outperform all existing methods.

5. In Lines 338-345, the claim that SUSI *"generalizes well and maintains scalability"* across model scales is premature and potentially overstated, as it is based solely on experiments with models up to 1.3B parameters. In the context of modern LLMs, a 1.3B model is considered relatively small, and the performance degradation patterns from pruning may differ significantly in much larger models (e.g., 7B, 30B, or 70B parameters) where sparsity techniques are of greater practical interest. To robustly support the claim of scalability, validation on these larger model scales is necessary.

6. In Section 4.3.3, the claim of robustness, based solely on similar learned masks across different random seeds, is weak. This consistency may simply indicate that the method converges to the same suboptimal local minimum rather than demonstrating true algorithmic stability. This interpretation is particularly plausible given that SUSI does not achieve state-of-the-art results; the masks found, while consistent, may not be optimal. A more convincing demonstration of robustness would involve testing stability under varying hyperparameters or data conditions, not just initialization.

7. For Figure 5, could you please explicitly state what the dark-colored parts represent in the context of the weight matrix? Furthermore, does this figure depict the entire linear layer, or is it a representative subset? It seems that it is just a representative subset. If so, how is it selected?

[1] Zhang, Yuxin, et al. "Dynamic sparse no training: Training-free fine-tuning for sparse llms." arXiv preprint arXiv:2310.08915 (2023).

[2] Zimmer, Max, et al. "Perp: Rethinking the prune-retrain paradigm in the era of llms." arXiv preprint arXiv:2312.15230 (2023).

[3] Boža, Vladimír. "Fast and effective weight update for pruned large language models." arXiv preprint arXiv:2401.02938 (2024).

[4] Zhao, Pengxiang, et al. "A convex-optimization-based layer-wise post-training pruner for large language models." arXiv preprint arXiv:2408.03728 (2024).

---

### Official Review · Reviewer_s9F5 · 2025-10-30

**Soundness:** 2
**Presentation:** 2
**Contribution:** 2
**Rating:** 2
**Confidence:** 4

**Summary:**

This paper proposes SUSI, a training-based semi-structured pruning method for Large Language Models. The approach leverages differentiable subset sampling via the Gumbel-Top-K trick to learn N:M sparsity masks. The core claimed contribution is a significant reduction in the number of trainable parameters compared to existing learnable masking approaches such as MaskLLM.

**Strengths:**

1.	The method significantly reduces the number of trainable parameters compared to MaskLLM—achieving up to 3.5x and 8.75x fewer parameters under the 2:8 and 4:8 sparsity patterns, respectively.
2.	Empirical robustness analysis of SUSI is included.

**Weaknesses:**

1.	Limited novelty: The core technique for ensuring differentiability—the Gumbel-Top-K trick—has been previously used in MaskLLM. Although SUSI introduces weighted reservoir sampling, the overall algorithmic framework does not represent a significant conceptual departure from existing differentiable pruning methods.
2.	SUSI shows competitive performance on some tasks, but its overall results on zero-shot accuracy benchmarks are mixed, and it does not clearly outperform strong baselines like SparseGPT, MaskLLM.
3.	The experiments are conducted only on OPT models (up to 1.3B parameters), which are relatively outdated and exhibit limited baseline performance. This raises concerns about whether the results would generalize to more capable or modern model families. The performance gaps between pruned and dense models are also more pronounced when SUSI is applied to newer architectures like Qwen2.5 and Llama3.2 in Appendix A.8.
4.	Although the training cost is reduced, the method still requires training a number of mask parameters on the same scale as the original model weights. This could become computationally prohibitive for models larger than 1.3B parameters (e.g., 7B+), which are commonly used in contemporary pruning studies. The paper does not demonstrate scalability beyond 1.3B.

**Questions:**

1.	The weighted reservoir sampling requires a loop over N steps which may greatly increasing the complexity of computational graph when N is large. Could the authors provide more detailed analysis on the memory and time costs for different values of N?
2.	To provide a more rigorous robustness analysis, we suggest running the experiment n times with different random seeds and calculating the Intersection-over-Union (IoU) for the n pruned subsets at each layer.

---

### Official Review · Reviewer_aYFY · 2025-10-31

**Soundness:** 2
**Presentation:** 2
**Contribution:** 2
**Rating:** 2
**Confidence:** 4

**Summary:**

The OASIS framework proposes a novel approach to improving post-training pruning of large language models (LLMs) by addressing the problem of calibration data selection. Traditional calibration data selection methods rely on simple heuristics, such as random sampling or entropy, which often result in suboptimal and inconsistent pruning outcomes. The authors point out that this inconsistency arises because the importance of calibration samples varies and is context-dependent (i.e., it depends on the specific model and pruning method). A key feature of OASIS is its end-to-end framework, which formulates calibration data selection as an optimization problem and solves it using a differentiable soft-mask proxy. This allows task-level gradients to be backpropagated to the calibration data, dynamically discovering the subset most beneficial for pruning. Experiments show that OASIS improves the performance of various state-of-the-art pruning methods, establishing a new standard for data-aware model compression.

**Strengths:**

1. Innovative method design: The paper combines weighted reservoir sampling and differentiable subset sampling to propose a novel semi-structured pruning strategy for efficiently learning N:M sparsity masks, demonstrating strong methodological innovation.
2. Significant reduction in trainable parameters: While maintaining model accuracy, SUSI substantially reduces the number of trainable parameters (up to 1.5× fewer) compared with MaskLLM, showing a good balance between efficiency and accuracy.
3. Comprehensive experimental design: The experiments cover OPT models of various scales (from 125M to 1.3B) and include analyses of ablation studies and training stability.

**Weaknesses:**

1. Limited practical improvement: SUSI only slightly outperforms Wanda on OPT-1.3B (46.48% vs. 46.32%), with negligible accuracy gain but much higher cost (2000 training steps vs. one forward pass). The practical advantage is unclear, especially for larger models.
2. Missing comparison with recent learnable-mask methods: The paper lacks comparison with newer approaches like ProxSparse, making it difficult to assess whether SUSI truly advances learnable mask pruning.
3. Limited experimental scope: The evaluation lacks comparisons with established baselines on standard models beyond OPT (such as Llama and Qwen), as well as validation on larger-scale architectures (e.g., DeepSeek), thus undermining the claims of generality.
4. No inference efficiency results: The paper reports accuracy but omits runtime metrics such as latency or FLOPs, so the actual acceleration effect remains unknown.
5. Overstated claims: Despite small accuracy gains and high cost, the paper repeatedly claims “significant improvement” and “reduced computation,” which are not well supported by data.

**Questions:**

1. It is recommended to include inference speed comparisons, evaluating the proposed method against baseline approaches such as Wanda and SparseGPT in terms of actual inference performance.
2. Add experiments on other models, such as LLaMA, Qwen, and DeepSeek, to verify whether SUSI can generalize and adapt to different architectures.
3. Clarify the applicability of SUSI beyond N:M sparsity: Can the proposed mask learning approach be extended to non-N:M sparsity patterns, and does it maintain scalability in such cases?

---

### Note · Authors · 2025-11-13

I have read and agree with the venue's withdrawal policy on behalf of myself and my co-authors.